# Primary Language in Relation to Knowledge of Diagnosis and Sun-Related Behaviors in Adults with Sun-Exacerbated Dermatoses

**DOI:** 10.3390/ijerph16193710

**Published:** 2019-10-02

**Authors:** Mayra B. C. Maymone, Stephen A. Wirya, Eric A. Secemsky, Neelam A. Vashi

**Affiliations:** 1Department of Dermatology, Boston University School of Medicine, Boston, MA 02118, USA; mayrabcm@gmail.com (M.B.C.M.); stephenakihiro@gmail.com (S.A.W.); 2Department of Medicine, Massachusetts General Hospital, Harvard Medical School, Boston, MA 02114, USA; ericsecemsky@gmail.com; 3US Department of Veteran Affairs, Boston Health Care System, Boston, MA 02118, USA

**Keywords:** health behavior, health knowledge, sun-exacerbated dermatosis, health outcomes

## Abstract

Objective: To evaluate how patients’ primary spoken language influences the understanding of their disorder and their subsequent sun-related behaviors. Methods: This was a cross-sectional study conducted between February 2015 and July 2016 in two outpatient dermatology clinics among 419 adults with a sun-exacerbated dermatosis. The primary outcome was a successful match between the patient-reported diagnosis on a survey and the dermatologist-determined diagnosis. Results: Of participants, 42% were native English speakers, and 68% did not know their diagnosis. Fewer non-native English speakers identified one risk factor for their condition (46% versus 54%, *p* < 0.01). A greater number of non-native English speakers were less familiar with medical terminology. Native English speakers were 2.5 times more likely to know their diagnosis compared to non-native speakers (adjusted odds (aOR) 2.5, 95% confidence interval, 1.32 to 4.5; *p* = 0.005). Additional factors associated with higher odds of knowing the diagnosis included: Higher education, sunscreen use, female gender, symptoms for 1–5 years, and diagnosis of melasma and postinflammatory hyperpigmentation (PIH). Conclusions: Knowledge of the diagnosis and understanding of factors that may influence skin disease may promote conscious sun behavior. Patients who knew that their diagnosis was sun-exacerbated had higher odds of wearing sunscreen.

## 1. Introduction

Sun-related behavior may be influenced by individual preferences and skin type, sociocultural background, and understanding of how intentions to engage a behavior can impact one’s health. Ultraviolet exposure is associated with increased risk of skin cancer, vitamin D production (thus bone health), and many dermatoses. The latter includes melasma, characterized as brown spots on face and postinflammatory hyperpigmentation (PIH), which presents as darkening of skin in region of previous trauma or inflammation. Both are worsened by sun exposure [1].

Defined as the capacity to obtain and understand health information to make appropriate health decisions, health literacy extends beyond educational literacy (reading and writing) to also involve cognitive and social skills that allow decision making [2,3,4,5]. Many factors affect health literacy acquisition by a non-native speaker, such as the quality of translation, complexity of the message, and cultural competency [6]. Better health literacy is associated with overall improved outcomes and practices that enhance health and reduce risk factors, such as sunscreen use to reduce exposure to ultraviolet radiation [5,7,8]. Correctly knowing and understanding one’s diagnosis is an important outcome of health-related interactions as it is necessary to making informed decisions. The important subject of health behavior among patients with sun-exacerbated dermatosis is understudied and is often presented in the literature independently from patients’ health outcomes. This study attempts to address this gap in the literature.

## 2. Patients and Methods

This was a cross-sectional study conducted in a convenience sample of 419 adults who sought dermatological care for a sun-exacerbated dermatosis at Boston Medical Center or East Boston Neighborhood Health Center between February of 2015 and July of 2016. Our sample was obtained from an academic institution and its affiliated community health center, which serves a broad range of individuals, many coming from lower socioeconomic means. This study was conducted in accordance with the ethical standards in research and was approved by the Boston University Institutional Review Board. 

Patients were eligible to participate if they were seeking dermatological care for sun-exacerbated dermatoses leading to skin hyperpigmentation in the two designated study settings; were older than 18; spoke English, Spanish, or Portuguese; and agreed to participate after verbal informed consent was obtained. Information was collected in two parts: A paper-based questionnaire in the participant’s preferred language (English, Spanish, or Portuguese) and a clinical assessment conducted by a trained dermatologist. All clinical interactions with those unable to communicate in English were performed with the help of a trained medical interpreter, either in person or via phone.

The questionnaire asked information about demographics including primary language, education, and race. Health behavior was assessed in two parts: General knowledge and use of protective measures. General knowledge was assessed in two concepts: Familiarity with five dermatological terms: Postinflammatory hyperpigmentation, melasma, freckles, and lentigo and risk factors including knowledge of having a medical diagnosis that was exacerbated with sun exposure. Protective measures targeted at reducing the risks for the skin condition were assessed as part of health behavior by a question concerning sunscreen use (yes/no).

Data Analysis

Patients’ knowledge of their diagnosis, a healthcare-related outcome, was measured as categorical variable (yes/no) by crossing the patient-reported diagnosis with the dermatologist-given diagnosis during the clinical assessment. Primary language was coded as a categorical variable based on the participants’ response with non-native English speakers defined as those who did not report English as their primary language. Familiarity with the five clinical terms was coded into five (yes/no) categorical variables, and identifying at least one risk factor for the patient dermatologist-given diagnosis was also measured as a categorical (yes/no) variable. A priori power analysis indicated that this study would require a sample size of at least 386 to detect a 0.5 difference in the odds ratio of knowing the diagnosis at a power of 80% in the two-tailed significant hypothesis.

Sample characteristics were described using frequency tables. Effects of primary language on aspects examining patients’ health behavior (identifying risk factors, familiarity with common clinical terms, and sunscreen use) in addition to race and education were described using frequency tables, and the chi-square test was utilized to find any statistically significant differences. A univariate and a multivariate logistic regression analysis were created to predict the odds of accurately knowing the diagnosis when the primary language was not English in reference to English. The multivariate model was adjusted for potential confounders identified based on clinical judgment and included: Gender, level of education (high school diploma or less versus college/graduate degree), sunscreen use (yes/no), race/ethnicity (Hispanic, African American, White, Asian), clinical diagnosis, and disease duration (defined as time since the patient developed symptoms (<1 year, 1–5 years, and >5 years). Race/ethnicity was subsequently excluded from the model as it was strongly correlated with language (*p* < 0.001). Data were transferred and managed using Research Electronic Data Capture (REDCap). Statistical analysis was performed using STATA/SE 13.0 (StataCorp LP, TX, USA). A two-sided *p* value of <0.05 was used to define significance. Analysis was performed on available data points as missing data were minimum and were assumed to be missing at random.

## 3. Results

A total sample of 419 patients agreed to participate out of 441 patients approached (95%). Most subjects were Hispanic or Latino (205/409 (50%)), female (373/419 (89%)), and between 35 and 44 years old (122/419 (29%)). Native English speakers comprised 42% (172/411) versus 58% (239/411) non-native English speakers. Of the study subjects, 50% (203/410) were single, 72% (285/396) were employed, and 45% (138/409) held a college or graduate degree. The two most common clinical diagnoses in this sample were melasma (167/419 (39%)) and PIH (138/419 (33%)). Overall, 68% (285/419) did not know their diagnosis (Table 1). 

In the unadjusted model, native speakers were about 3.1 times (95% CI, 1.98–4.69; *p ≤* 0.01) more likely to know their diagnosis than non-native speakers. In the adjusted model, the odds of knowing the diagnosis among native speakers were 2.5 times the odds among non-native speakers (1.32–4.5; *p* = 0.005). Other factors that were associated with the higher odds of knowing the diagnosis included higher level of education (adjusted odds—aOR, 2.23; 1.2 to 4.1; *p* = 0.01), using sunscreen (aOR, 2.43; 1.25 to 4.71; *p* = 0.008), female gender (aOR, 4.43; 1.4 to14; *p* = 0.011), having the diagnosis for 1–5 years (aOR, 2.85; 1.1 to 7.2; *p* = 0.028), a diagnosis of melasma (aOR, 4.2; 2 to 8.7; *p* < 0.001), and a diagnosis of PIH (aOR, 3.4; 1.6 to 7.5; *p* = 0.02) (Table 2).

The majority of those who identified as Whites (40/58 (70%)), African Americans (79/118 (69%)), or Asians (19/28 (68%)) reported English as their primary language, whereas only 31 of the 205 who identified as Hispanics (15%) reported English as their primary language. More non-native speakers had a lower education level with high school or less comprising 80% (176/278) compared to 20% (45/278) of native English speakers (*p* < 0.01), lower knowledge of risk factors associated with their condition with 46% (104/169) identifying at least one risk factor compared to 54% (123/169) of native speakers (*p* < 0.01), and less familiarity with medical terms than native speakers overall (PIH: 23% (27/117) versus 77% (89/117), *p* < 0.01; melasma: 48% (51/108) versus 52% (55/108), *p* = 0.02; solar lentigo: 44% (80/185) versus 56% (102/185), *p* < 0.01; freckles: 48% (125/266) versus 52% (137/266), *p* = 0.003 (Table 3).

## 4. Discussion

Our study highlights important considerations regarding health behavior, especially among non-native English speakers. Fewer non-native English speakers knew their diagnosis (52/239 (22%)) compared to native English speakers (79/172 (46%)) (Table 3), despite 289 of 362 (82%) having symptoms for longer than one year (Table 2). This may be explained by either a delay in seeking care or impaired health behavior due to overall poor communication. Both conclusions signify important issues to be explored in further research, and targeted interventions aiming to increase health education and improving in-clinic communication. 

Most patients in this sample were females, which could be explained by a gender-related discrepancy in health-seeking behavior for skin-related issues. This observation is in accord with similar reported trends in dermatology research, in which females were overly represented in the sample [9] or had higher odds of seeking care than males [10]. This discrepancy could point to a health behavior gap in the male population centered around who should seek dermatological care, when, and for what reasons. 

The language barrier in this sample puts the majority of non-native English speakers at a statistically significant disadvantage, with less than half of non-native speakers having a college or graduate degree compared to native speakers. Education is a significant factor affecting health behavior skills [2,6,10]. Many concepts highlighted in various definitions of health behavior [2,6] rely heavily on cognitive and social skills, a relatively fixed intrinsic skill set for each individual [6]. Attaining a higher level of education might be directly linked to some of these intrinsic skills or could provide the tools and/or the environment necessary to promote mastering of these skills. The significantly lower success in identifying at least one risk factor for the skin condition in non-native versus native speakers and the much lower percentage of those who had prior knowledge of any of the common clinical diagnoses can be explained along similar lines and highlights the disadvantages of non-native speakers in navigating and understanding the healthcare system, compromising their ability to make informed decisions about their health.

The regression model speaks further to the disadvantage among non-native speakers. In the adjusted model, native speakers remained 2.5 times more likely to know their diagnosis even when education, gender, duration of the disease are controlled for diagnosis. However, there may also be other factors involved in this relationship, including the proficiency and quality of medical interpretation during interactions with healthcare providers. Although useful and mandatory in many settings, the translator might create an interaction barrier resulting in missed opportunities for the patient and the provider to connect beyond words with social smiles and gestures of respectful care [6]. This complexity heightens when the translator is a family member, which might occur in a clinical setting if interpretation services are unavailable or refused [10]. Table 4 contains suggestions to improve clinic visit communication.

Our study reported that those who used sunscreen were 2.4 times more likely to know their diagnosis compared to those who did not. Greater knowledge about underlying medical condition may improve health behavior. The strengths of this study are the large sample size and data collection with professional translation of the questionnaire to three different languages.

Study limitations include lack of standardized measures that address all aspects of health behavior, which involves patient factors as well as health-system-related factors and inability to control for quality of translation in prior interactions with healthcare providers. Our study is limited by having a predominantly female population and our single-center site, limiting the generalizability of our study.

## 5. Conclusions

In conclusion, patients who knew that their diagnosis was sun-exacerbated had higher odds of wearing sunscreen. Therefore, knowledge of the diagnosis and understanding of factors that may influence skin disease may promote conscious sun-protective behavior. Moreover, language was an important barrier to achieving successful health behavior and better health outcomes in the patient. Developing interventions directed at health education, cultural competency, and improving in-clinic communication by increasing patients’ knowledge and decreasing the complexity of language used by healthcare providers and dermatologists is vital to ensure quality of care and improve health outcomes.

## Figures and Tables

**Table 1 ijerph-16-03710-t001:** Basic demographic characteristic of the study population (*n* = 419) ^a^.

Characteristic	*n* (%)
**Age**	
18–24	30 (07.16)
25–34	116 (27.68)
35–44	122 (29.12)
45–54	84 (20.05)
≥55	67 (15.99)
**Gender**	
Female	373 (89.02)
Male	46 (10.98)
**Race/ethnicity (*n* = 409) ^a^**	
White	58 (14.18)
African American	118 (28.98)
Hispanic or Latino	205 (50.01)
Asian	28 (6.84)
**Education (*n* = 409)**	
Lower/elementary school	37 (9.05)
Middle school	55 (13.45)
High school	134 (32.76)
College	108 (26.41)
Graduate school	75 (18.33)
**Diagnosis**	
Melasma	167 (38.98)
Postinflammatory hyperpigmentation (PIH)	138 (33.41)
Other	114 (27.61)
**Primary Language (*n* = 411)**	
English	172 (41.85)
Non-English	239 (58.15)
**Marital status (*n* = 410)**	
Married	152 (37.07)
Single	203 (49.51)
Other	55 (13.41)
**Currently employed (*n* = 396)**	
No	111 (28.03)
Yes	285 (71.97)
**Knows the diagnosis**	
No	285 (68.02)
Yes	134 (31.98)

^a^ Denominators may vary due to missing data.

**Table 2 ijerph-16-03710-t002:** Factors associated with knowing the diagnosis among patients with sun-exacerbated dermatosis.

Dependent Variables	Knows the Diagnosis ^a^		Odds Ratio	*p* Value	Confidence Interval	
	Yes	No				
**Primary language**						
Other languages	52 (21.76)	187 (78.24)	--	--	--	--
Native English speakers	79 (45.93)	93 (54.07)	2.5	0.005	1.32	4.5
**Level of education**						
High school diploma or less	51 (22.57)	175 (77.43)	--	--	--	--
College/graduate degree	81 (44.26)	102 (55.74)	2.23	0.010	1.20	4.13
**Sunscreen use**						
No	26 (20.47)	101 (79.53)	--	--	--	--
Yes	171 (61.51)	107 (38.49)	2.43	0.008	1.25	4.71
**Gender**						
Male	7 (15.22)	39(84.78)	--	--	--	--
Female	127 (34.05)	246 (65.95)	4.43	0.011	1.4	14
**Duration of the disease**						
<1 year	57 (89.06)	7 (10.94)	--	--	--	--
1–5 years	89 (65.44)	47 (34.56)	2.85	0.028	1.1	7.2
>5 years	103 (63.58)	59 (36.42)	2.33	0.07	.92	5.9
**Clinical Diagnosis**						
Other hyperpigmentation disorders	81 (81.00)	19 (19.00)	--	--	--	--
Melasma	102 (63.35)	59 (36.65)	4.2	0.001	2.0	8.7
PIH	82 (59.42)	56 (40.58)	3.4	0.002	1.6	7.5

^a^ Denominators might differ due to missing data.

**Table 3 ijerph-16-03710-t003:** Clinical Characteristic of the study population.

Attribute	*n* (%)	Primary Language		*p* Value ^c^
		English *n* (%)	Non-English *n* (%)	
**Race (*n* =409) ^a,b^**				
White	58 (14.18%)	40 (70.18)	17(29.82)	<0.01
African American	118 (28.85 %)	79 (68.70)	36 (31.30)	
Asian	28 (6.85%)	19 (67.86)	9 (32.14)	
Hispanic/Latino	205 (50.12)	31 (15.42)	170 (84.58)	
**Education (*n* = 405)**				
High school or less	278 (66.35)	45 (20.36)	176 (79.64)	<0.01
College/graduate degree	127 (30.31)	124 (68.89)	56 (31.11)	
**Duration since symptoms**				
<1 year	64 (17.68)	16 (25.00)	48 (75.00)	
1–5 years	136 (37.57)	49 (37.12)	83 (62.88)	<0.01
>5 years	162 (44.75)	89 (55.62)	71 (44.38)	
**Attributed skin condition to at least one correct risk factor (*n* = 400)**				
Yes	169 (42.25)	123 (54.19)	104 (45.81)	<0.01
No	231 (57.75)	36 (23.68)	116 (76.32)	
**Heard of PIH before (*n* = 407)**				
Yes	117 (28.75)	89 (76.72)	27 (23.28)	<0.01
No	290 (71.25)	80 (28.27)	203(71.73)	
**Heard of melasma before (*n* = 410)**				
Yes	108 (26.34)	55 (51.89)	51 (48.11)	0.020
No	302 (73.66)	115 (38.85)	181(61.15)	
**Heard of solar lentigo before (*n* = 403)**				
Yes	185 (45.91)	102 (56.04)	80 (43.96)	<0.01
No	218 (54.09)	150 (69.77)	65 (30.23)	
**Heard of freckles before (*n* = 409)**				
Yes	266 (65.04)	137 (52.29)	125 (47.71)	0.003
No	143 (34.96)	30 (21.58)	109 (78.42)	

^a^ Denominators might differ due to missing data. ^b^ Denominators apply to the relationship between attributes and primary spoken language. ^c^ Generated through chi-square test.

**Table 4 ijerph-16-03710-t004:** List of suggestions to improve clinic visit communication [11,12,13].

• Bilingual patient navigator.
• One-touch telephone interpreter available.
• Use trained medical professional interpreters.
• When you have the interpreter on the line, make sure to introduce yourself and briefly explain the patient history.
• Ask the interpreter to introduce themselves to the patient, and let the patient know that he/she can ask anything.
• While speaking to the interpreter, communicate looking directly at the patient.
• Speak in short sentences and ask for clarification when necessary.
• Try not to interrupt the interpreter, and if the interpreter or the patient seem confused, always ask for clarification.
• Before finishing the medical encounter, ask the patient if he/she has any questions and review the encounter including understanding of the diagnosis and medical therapy recommended.

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
