# Peer review of "Primary Language in Relation to Knowledge of Diagnosis and Sun-Related Behaviors in Adults with Sun-Exacerbated Dermatoses"

_ijerph, 2019, doi:10.3390/ijerph16193710_

Round 1
Reviewer 1 Report
This paper addresses an interesting and potentially important topic. However, the methods and writing need work. The biggest problem is that this study did not assess health literacy.
Title: Does not capture the nature of the study very well and is not well-written.
Abstract
1. Please do not talk about “effects” when the study is correlational.
2. Change “less of” to “fewer”. “More were less” is awkward.
3. Suggest reorganizing results so that sentences about knowledge of diagnosis are first and together.
4. What is the comparator for symptoms for 1-5 years?
5. Write out abbreviations on first appearance.
Intro: Since this is not a derm journal, and many of your readers will be non-MDs, more background info about medical terms are needed.
Methods
1. Informed consent is a process, and patients sign informed consent forms/documents. “Approved an informed consent” is inaccurate and non-specific.
2. Somewhere, the demographics of the clinic or geographic area should be described so that readers can see how representative the sample was. E.g., The sample seems somewhat young.
3. Health literacy is not the same as health knowledge or health behavior. The items, responses, scoring, psychometrics, and citations of all the measures are needed.
4. Why were potentially continuous or ordinal variables coded as dichotomous? https://link.springer.com/article/10.1007/s12529-019-09790-7?wt_mc=alerts.TOCjournals&utm_source=toc&utm_medium=email&utm_campaign=toc_12529_26_4
Results
1. It seems like it might make sense to consider Latinos vs. others instead of or in addition to non-English vs. others. The non-English speaking sample is primarily Latino.
2. Some of the race and education data are included in both Table 1 and 2.
3. Table 3: Be consistent about noting the reference group for all variables or not. p-values may not be necessary with CIs. If they are to be included, p-value should be bolded.
Discussion
1. Is there anything more anecdotally that can be said about lay interpreters? I would assume they may not know terms such as melasma in Spanish, so these patients may have a very hard time remembering their diagnoses. These issues should be emphasized.
2. Underrepresentation of men should be listed as a limitation.
3. This sentence is speculative and not well-justified - “This may be explained by either a delay in seeking care or impaired health literacy due to overall poor communication.”
4. The sample may be more female because they are more likely to participate in research, more likely to seek derm care, and/or more likely to be interested in conditions related to appearance.
5. Sentences should not end in prepositions (line 151, “controlled for”).
6. These statements are not drawn from study data - “This provides evidence that health literacy concepts are not independent of each other, so that when patients understand their skin disease and the factor that exacerbate their condition they are more likely to adopt and modify their sun protective behavior.”
7. “In conclusion, patients who knew that their diagnosis was sun exacerbated had higher odds of wearing sunscreen.” – It was not previously stated that whether patients “knew their diagnosis was sun exacerbated” was measured.
Author Response
Dear Reviewer’s
Thank you for your review of our manuscript and return with major revisions. Please find below reviewers’ comments from our submission. We have included point-by-point responses to each comment. In addition, changes to the main text can be found highlighted in the manuscript file. We appreciate the comments and are thankful for the valuable contribution. Thank you again for your kind consideration.
Reviewer 1
This paper addresses an interesting and potentially important topic. However, the methods and writing need work. The biggest problem is that this study did not assess health literacy.
Title: Does not capture the nature of the study very well and is not well-written.
Thank you for this important comment. We have changed the study title to better reflect the study nature.
Abstract
Please do not talk about “effects” when the study is correlational.
This is now corrected to the more appropriate terminology.
Change “less of” to “fewer”. “More were less” is awkward.
Thank you. This is now corrected.
Suggest reorganizing results so that sentences about knowledge of diagnosis are first and together.
Thank you. The paragraphs have now been reorganized as suggested.
What is the comparator for symptoms for 1-5 years?
Comparator groups were < 1 year and > 5 years.
Write out abbreviations on first appearance.
Abbreviations have now been written out on first appearance.
Intro: Since this is not a derm journal, and many of your readers will be non-MDs, more background info about medical terms are needed.
The introduction was rewritten to be more pertinent for non-MDs.
Methods
Informed consent is a process, and patients sign informed consent forms/documents. “Approved an informed consent” is inaccurate and non-specific.
Thank you for bringing this to our attention. The sentence is now fixed to reflect the proper process of obtaining informed consent.
Somewhere, the demographics of the clinic or geographic area should be described so that readers can see how representative the sample was. E.g., The sample seems somewhat young.
We have now included information in regards to the demographics of our clinic/geographic area. We serve a tertiary referral center which is quite diverse. We cannot find a reason as to why our population appears younger; therefore, we have also included this in our limitations.
Health literacy is not the same as health knowledge or health behavior. The items, responses, scoring, psychometrics, and citations of all the measures are needed.
We agree with reviewer. As such, we have changed our title to better reflect our study and to not overstep our conclusions and inferences. This was a pilot study in which we did not use a validated questionnaire.
Why were potentially continuous or ordinal variables coded as dichotomous?
The ordinal variables were transformed to yes/no to better answer the research questions of the study.
Results
It seems like it might make sense to consider Latinos vs. others instead of or in addition to non-English vs. others. The non-English speaking sample is primarily Latino.
Thank you for this thought. Given our experience with a large Hispanic population, we felt that Latinos versus Not was not an appropriate way to group patients as there are large variations to English speaking ability.
Some of the race and education data are included in both Table 1 and 2.
Race was removed from table 2.
Table 3: Be consistent about noting the reference group for all variables or not. p-values may not be necessary with CIs. If they are to be included, p-value should be bolded.
We have removed the reference group, and we bolded the p-values.
Discussion
Is there anything more anecdotally that can be said about lay interpreters? I would assume they may not know terms such as melasma in Spanish, so these patients may have a very hard time remembering their diagnoses. These issues should be emphasized.
Boston Medical Center and East Boston clinic provide service of professional medical/healthcare interpreters to help with translation during a visit encounter usually over the phone and sometimes in person. The use of lay interpreters, or bilingual family members is not encouraged nor a common practice.
Underrepresentation of men should be listed as a limitation.
Thank for this important point, this is now listed in the study limitations.
This sentence is speculative and not well-justified - “This may be explained by either a delay in seeking care or impaired health literacy due to overall poor communication.”
The sentence was removed.
The sample may be more female because they are more likely to participate in research, more likely to seek derm care, and/or more likely to be interested in conditions related to appearance.
We agree, and we added this to our limitations.
Sentences should not end in prepositions (line 151, “controlled for”).
Thank you, and the sentence was rephrased.
These statements are not drawn from study data - “This provides evidence that health literacy concepts are not independent of each other, so that when patients understand their skin disease and the factor that exacerbate their condition they are more likely to adopt and modify their sun protective behavior.”
We agree and have removed it from the manuscript.
“In conclusion, patients who knew that their diagnosis was sun exacerbated had higher odds of wearing sunscreen.” – It was not previously stated that whether patients “knew their diagnosis was sun exacerbated” was measured.
Thank you for bringing this to our attention. A sentence was included in the methods section.
Thank you for your kind consideration!
Reviewer 2 Report
Overall, a very interesting paper outlining the the interaction between healthy literacy, language and patient behaviour. I have listed below points which I believe will strengthen the manuscript.
The title is somewhat misleading - I would suggest replacing the word sunscreen with knowledge of diagnosis.
It is not clear precisely what the objective/aim of the study is. It is not stated clearly in the body of the manuscript
Page 1 Line 33 and 34 - Please revise this sentance to be grammatically correct. Please outline which health outcomes UV exposure is directly related to.
The introduction would benefit from a brief explanation of common sun exacerbated dermatosis.
Within the abstract the objective is listed as being to understand the effect of health literacy and language on patient-related outcome. I think it would be a better representation to state that the outcome of interest is knowlegde of diagnosis as stated within the methods section.
Page 4 Line 108/109 - Please justify why the unadjusted data is not shown and the precise Odds Ratio estimate.
Section 4. Discussion Paragraph begining Line 145 - Please outline whether there are other conditions or instances where non-native speakers have been found to have lower odds of knowing their diagnosis.
Line 157 within the Discussion requires a reference.
At present it is unclear if the authors are stating that sunscreen use predicted knowledge of diagnosis or if knowledge of diagnosis predicted sunscreen use. Please clarify.
Provide evidence and discussion on interventions which could improve in-clinic communication to overcome language and health literacy barriers.
The authors do not attempt to weigh up the strengths and limitations of their study.
Author Response
Reviewer 2
Overall, a very interesting paper outlining the interaction between healthy literacy, language and patient behaviour. I have listed below points which I believe will strengthen the manuscript.
The title is somewhat misleading - I would suggest replacing the word sunscreen with knowledge of diagnosis.
Thank you for the suggestion. We have changed the title accordingly.
It is not clear precisely what the objective/aim of the study is. It is not stated clearly in the body of the manuscript.
We have reworded the study objective.
Page 1 Line 33 and 34 - Please revise this sentance to be grammatically correct. Please outline which health outcomes UV exposure is directly related to.
Thank you for this comment. We have rephrased this sentence.
The introduction would benefit from a brief explanation of common sun exacerbated dermatosis.
Thank you, we have addressed this comment to include a brief explanation of common sun exacerbated dermatoses.
Within the abstract the objective is listed as being to understand the effect of health literacy and language on patient-related outcome. I think it would be a better representation to state that the outcome of interest is knowlegde of diagnosis as stated within the methods section.
Thank you for this comment, we have added it to the objectives.
Page 4 Line 108/109 - Please justify why the unadjusted data is not shown and the precise Odds Ratio estimate.
We hoped to present our worked in a concise manner, including only 3 tables (which we have now already expanded to 4 based on another reviewer comment). However, we would be happy to add one more if the reviewer desires. We have removed the sentence mentioning the unadjusted model.
Section 4. Discussion Paragraph begining Line 145 - Please outline whether there are other conditions or instances where non-native speakers have been found to have lower odds of knowing their diagnosis.
Thank you for this important point. Yes, having a shorter disease duration, lower level of education had lower odds of knowing the diagnosis.
Line 157 within the Discussion requires a reference.
Thank you. We have included the reference.
At present it is unclear if the authors are stating that sunscreen use predicted knowledge of diagnosis or if knowledge of diagnosis predicted sunscreen use. Please clarify.
We apologize for the confusion. Those who knew that their diagnosis was sun exacerbated had higher odds of using sunscreen.
Provide evidence and discussion on interventions which could improve in-clinic communication to overcome language and health literacy barriers.
Thank you for the suggestion. We have created a table 4 with suggestions on how to improve clinic communication.
The authors do not attempt to weigh up the strengths and limitations of their study.
This is a very important point. We apologize for not including this and have now included strengths/limitations.
Thank you for your kind consideration!